# SVvalidation: A long-read-based validation method for genomic structural variation

Yan Zheng [ID]*, Xuequn Shang*

School of Computer Science, Northwestern Polytechnical University, Xi'an, China

* yan.zheng@nwpu-bioinformatics.com (YZ); shang@nwpu.edu.cn (XS)

## Abstract

Although various methods have been developed to detect structural variations (SVs) in genomic sequences, few are used to validate these results. Several commonly used SV callers produce many false positive SVs, and existing validation methods are not accurate enough. Therefore, a highly efficient and accurate validation method is essential. In response, we propose SVvalidation—a new method that uses long-read sequencing data for validating SVs with higher accuracy and efficiency. Compared to existing methods, SVvalidation performs better in validating SVs in repeat regions and can determine the homozygosity or heterozygosity of an SV. Additionally, SVvalidation offers the highest recall, precision, and F1-score (improving by 7-16%) across all datasets. Moreover, SVvalidation is suitable for different types of SVs. The program is available at https://github.com/nwpuzhengyan/SVvalidation.

**Data Availability Statement:** All relevant data are within the paper and its Supporting information files.

**Funding:** The author(s) received no specific funding for this work.

## Introduction

Structural Variations (SVs) [1] refer to large-scale mutations in a genome, including deletions, insertions, duplications, inversions, and translocations, with a length of at least 50 base pairs. Although less frequent than SNPs and small indels, recent research has shown that SVs are crucial to the development of many genetic disorders such as cancer, autism, and Alzheimer's disease [2–4]. SVs also have significant impacts on evolution [5, 6], gene expression [7], and phenotype [8, 9]. In the past decades, numerous SV caller methods have been developed to detect SVs in whole genomes. Earlier methods such as Lumpy [10], Delly [11], Manta [12], Pindel [13], Gustaf [14], and SurVIndel [15] used short read sequencing data (read length of 100–150 bp) and mostly relied on read depths, discordant read pairs, split read alignments, local assembly or combinations of these techniques. However, with the development of long-read sequencing data from technologies like Pacific Bioscience [16] and Oxford Nanopore [17], an increasing number of long-read based SV callers are now available, including cuteSV [18], Sniffles [19], Sniffles2 [20], NanoSV [21], picky [22], SVIM [23], PBHoney [24], PBSV [25], SVDSS [26], DeBreak [27], SVsearcher [28] and SVision [29]. Unfortunately, we have observed that current SV callers (especially those based on short-read data) suffer from a high rate of false positive SVs, and there is not yet a reliable validation method available. Therefore, we propose a new validation method to filter out erroneous SV calls effectively.

**Competing interests:** The authors have declared that no competing interests exist.

Although short-read sequencing data are essential in significant genomics studies, such as the 1000 Genomes Project. However, due to their read length limitations, they have a high misalignment rate which hampers their use in validating structural variations (SVs). The presence of many repeat regions in the genome exacerbates the problem by leading to wrong alignments. On the other hand, long-read sequencing data have longer read lengths, making them easily align with accuracy. Although long-read sequencing has a slightly higher error rate (approximately 5%-20%) [30], it still validates SVs better than short-read sequencing data. Additionally, the PacBio technology [16] generates accurate long-read sequencing data (HiFi data) with a sequencing error rate below 1%. HiFi reads have lower misalign rates compared to other sequencing types. Based on these factors, we decided to use long-read sequencing data to validate SVs.

Although researchers have developed numerous methods for calling structural variations (SVs), few methods exist for validating their results. Only one method dedicated to validating SVs, called vapor [31], has been identified. However, we have found that the vapor method is not sufficient for this task. The method employs a fixed-size window (k-mer) and slides it with a 1-bp step through each read to identify positions where the read sequence and reference are identical. Unfortunately, subreads created by this window are difficult to align to the correct position in repeat regions. This leads to poor performance of the vapor method for detecting SVs in these regions. Additionally, the method struggles to filter out false positive SVs. We created a list of false SVs, which includes those with an error length and error rate greater than 50%; vapor outputs many false SVs. As a result, the sensitivity and accuracy of the vapor method are inadequate.

Based on the above observations, our paper introduces a novel method named SVvalidation, which can effectively validate Structural Variations (SVs) including those present in repeat regions and those having error length. This new method can assist others in validating the credibility of SVs and determining the accuracy of different caller outcomes. When compared to current validation methods, SVvalidation exhibits an improved F1-score ranging from 7% to 16%. Additionally, it is noteworthy that SVvalidation can accurately validate all types of SVs.

## Materials and methods

### Details of the data

There are two prevailing long-read technologies in use today—PacBio (Pacific Biosciences) [16] and ONT (Oxford Nanopore Technologies) [17]. The key advantage of PacBio data is its superior accuracy rate, whereas the primary advantage of ONT data is its longer read length. Our research team obtained both types of long-read data and applied validation methods to each. Ultimately, we compared the results from the two datasets in order to identify differences between them. To test the performance of different methods, we selected the HG002 and CHM13 as research data since we can get a very high confident benchmark for the two samples. For HG002, we can directly get a high confident benchmark-HG002_SVs_Tier1_v0.6, which is currently widely accepted by researchers and can be obtained from the GIAB website. As for CHM13, its near-complete homozygosity allows us to treat it as a haploid human genome. By comparing its assembly with the reference using Assemblytics [32], we can obtain a high-confidence benchmark for CHM13. However, for other samples where heterozygous SVs are present (one from the mother and one from the father), achieving nearly 100% accuracy in benchmarking is challenging. Further details regarding the two samples are provided in the following subsections.

**HG002.** We evaluated the performance of SVvalidation and vapor on HG002 genome mapped to GRCh37 genome. The Genome in a Bottle (GIAB) consortium has developed an integration pipeline that combines sequencing data generated from various technologies to create a comprehensive list of structural variants (SVs) named HG002_SVs_Tier1_v0.6 for use when benchmarking SV callers [33]. This SV list is publicly available at https://ftp-trace.ncbi. nlm.nih.gov/giab/ftp/data/AshkenazimTrio/analysis/NIST_SVs_Integration_v0.6/HG002_ SVs_Tier1_v0.6.vcf.gz. We used only pass-type SVs from this list as benchmarks, which have very high confidence. There were 5463 deletions (DELs) and 7279 insertions (INSs) in our benchmark dataset, all of which are considered 100% true SVs and have correct lengths greater than 50 base pairs. The Table S1 in S1 File provides a link to the long-read sequencing data (HG002), which we downloaded and aligned with the GRCh37 reference genome. Later, we applied both SVvalidation and vapor tools to the alignment file and compared their performances. The section 1 in S1 File contains detailed Linux commands for our analysis.

**CHM13.** Except for the HG002, it is difficult for us to establish a benchmark with high confidence. However, when compared to HG002, CHM13 displays almost complete homozygosity and can thus be considered a haploid human genome [34]. It has a high-quality assembly as the reference genome. Assemblytics [32] is a method for calling SVs from the assembly and the recall rate is about 90.9 − 99.9% and the false positive rate is about 0.29 − 0.40% in simulated data. Therefore, we employed this method to compare CHM13 and GRCh38 to obtain a benchmark of SVs for CHM13 on GRCh38. The accuracy of this benchmark was also close to 100%, with 3647 deletions and 6457 insertions identified. The Table S1 in S1 File contains the link to the long-read sequencing data of HG002 that we downloaded and aligned into the GRCh38 reference genome. We ran various validation methods and compared their performance as well, with detailed Linux commands shown in section 1 of the S1 File.

## Overview of SVvalidation

Based on our observations (as described in the first two subsections of the Results section), we have discovered that vapor is ineffective in repeat regions and incapable of filtering out SVs with erroneous length. Consequently, we propose a novel method called SVvalidation which validates SVs with higher accuracy, determines the correct length of SVs, and outputs the precise SVs compared to existing methods. The input for SVvalidation includes a long-read BAM file that has been sorted, as well as a BED file containing SVs lists. The SVvalidation process entails three primary steps: (1)parameter estimation, (2) identifying support reads, and (3) validating and outputting correct SVs. You can refer to Fig 1 for a visual representation of these steps. The following sections detail each step more thoroughly.

**Parameter estimation.** To align long reads with the reference genome, we currently use an aligner and sort the resulting alignments using samtools. This yields a sorted bam file, which along with a bed file of SVs lists, serves as the input data for SVvalidation. Before validating the SVs, we process the bam file and estimate its parameters. Specifically, we randomly select 1000 nodes from the reference and calculate their average coverage. The formula we use is as follows:

$$average\_coverage_{bamfile} = \frac{1}{1000} \sum_{i=1}^{1000} coverage_i$$

In the genome, certain regions exhibit significantly higher coverage levels than the average, sometimes exceeding dozens of times. This suggests that reads from the other regions are aligned into these regions due to high similarity. It becomes challenging to distinguish which reads correspond to which specific region. Therefore, validating structural variations (SVs) in

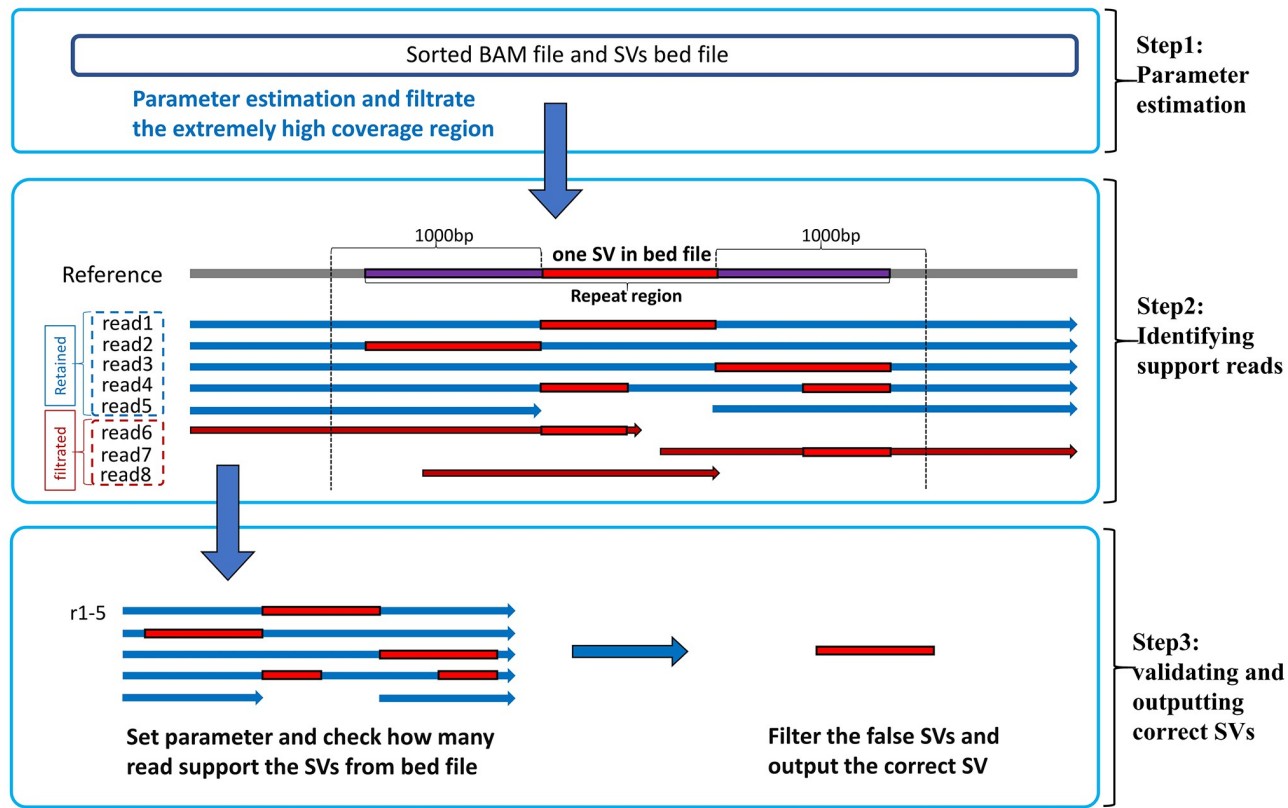

**Fig 1. The overview of SVvalidation.** There are three main steps in SVvalidation. (1) parameter estimation, (2) Identifying support reads, (3) validating and outputting correct SVs.

these regions poses difficulties. However, the SVs in these abnormally high coverage regions are only a small fraction of the actual number. In the CHM13 benchmark, only 0.5% of DEL and 1% of INS are in the regions where the coverage exceeds five times the average coverage. This has little effect on the results of the validation. Therefore, we filter out SVs occurring in regions with coverage exceeding five times the average.

**Identifying support reads.** As for the remaining SVs, two methods are utilized to identify their support reads. The first method involves checking the cigar string of reads, similar to read1–4 in step 2 of Fig 1. The second method involves checking split reads, as depicted by read5 in step 2 of Fig 1. Typically, short SVs rely mainly on read1 in Fig 1 as their support reads, whereas long SVs mostly use read5 in Fig 1 as their support reads.

Before identifying support reads, we first screened the reads in this region by retaining only reads with MAPQ $\geq$ 20. If this region is a repeat region, it may be challenging for some reads to accurately represent SVs (just like the read4, read6–8 in step2 of Fig 1). To filter out incorrect representation of SVs, we exclude reads 6–8 because they can only demonstrate an incorrect SV. However, although read4 divides the correct SVs into two parts, merging these parts produces the correct SVs. Therefore, we set a parameter, flank_len (default 1000bp). If a read cannot go through the region (SV_start_pos-flank_len, SV_end_pos+flank_len), we consider the read invalid and filter it out (the red reads in step2). In long-read data, due to noise interference, one SV may appear as multiple SVs in alignments (as illustrated in Fig 2), leading to incorrect validation results. We propose a solution to this problem: if there are multiple SVs in

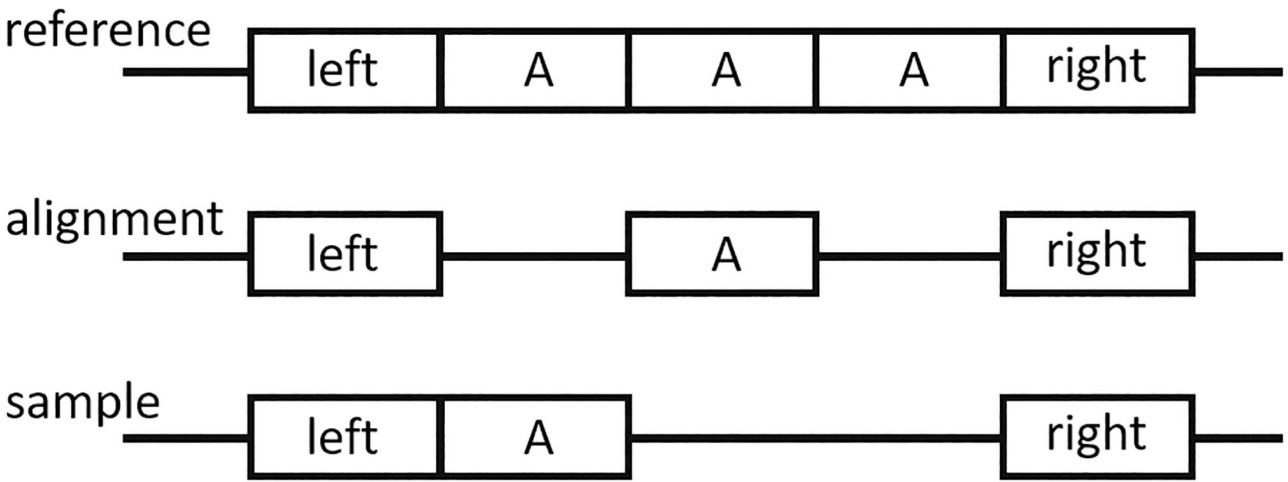

**Fig 2. From the figure, it is apparent that the sample has two deleted A regions in comparison to the reference.** However, given the influence of long-read data noise, we may observe the alignment at the center of the figure. For this type of alignment, we intend to combine multiple structural variants (SVs) in the repeat region and use their total SV length to validate them.

the same region, we calculate the total SV length and determine whether the read supports any of the SVs listed in the bed file.

Then We check the cigar string and split read for the retained reads (the blue reads in step 2) to determine whether they support different types of SVs, as illustrated in Fig 3. The specific procedures are described in the following sections for further clarity.

**1.Check cigar string**. For the DEL or INS from the bed file (labeled as $indel_i$), it is easy to judge whether this read is a support read. We check the cigar string and record the DEL or INS (length $\geq$ 30) in the region ($indel_i$_start_pos-flank_len, $indel_i$_end_pos+flank_len). Next, we calculate the total indel length of the above region and compare it with the length of $indel_i$. If abs(total_indel_length-$indel_i$_length) $\leq$ distance_support, we think this read supports $indel_i$. The distance_support formula is as follows:

$$distance\_support = 0.2*indel_i\_length + 2000/indel_i$$

For the INV from the bed file (labeled as $INV_i$), We can't check out INV directly from the cigar string. Hence, we first check the region ($INV_i$_start_pos, $INV_i$_end_pos) and record the noise number (noise includes indels and mismatch, this noise is labeled as noise1). Next, we take the sequence ($INV_i$_start_pos, $INV_i$_end_pos) from the reference and reverse it. Finally, we compare the reversed sequence with the read and record the noise number (labeled as noise2). If the read can meet the following criteria, we consider the read supports $INV_i$.

1. noise1/$INV_i$_length $\geq$ 0.3.

2. noise2/$INV_i$_length $\leq$ 0.2.

These two criteria mean that the read has poor mapping quality in the previous reference, but better mapping quality in the reversed reference.

**2.Check split read**. For large SVs, it is very hard to find SV signatures directly from the cigar string. In such cases, reads located near these long SVs often split into multiple alignments resulting in split reads. To identify potential SVs, all reads adjacent to it with a primary

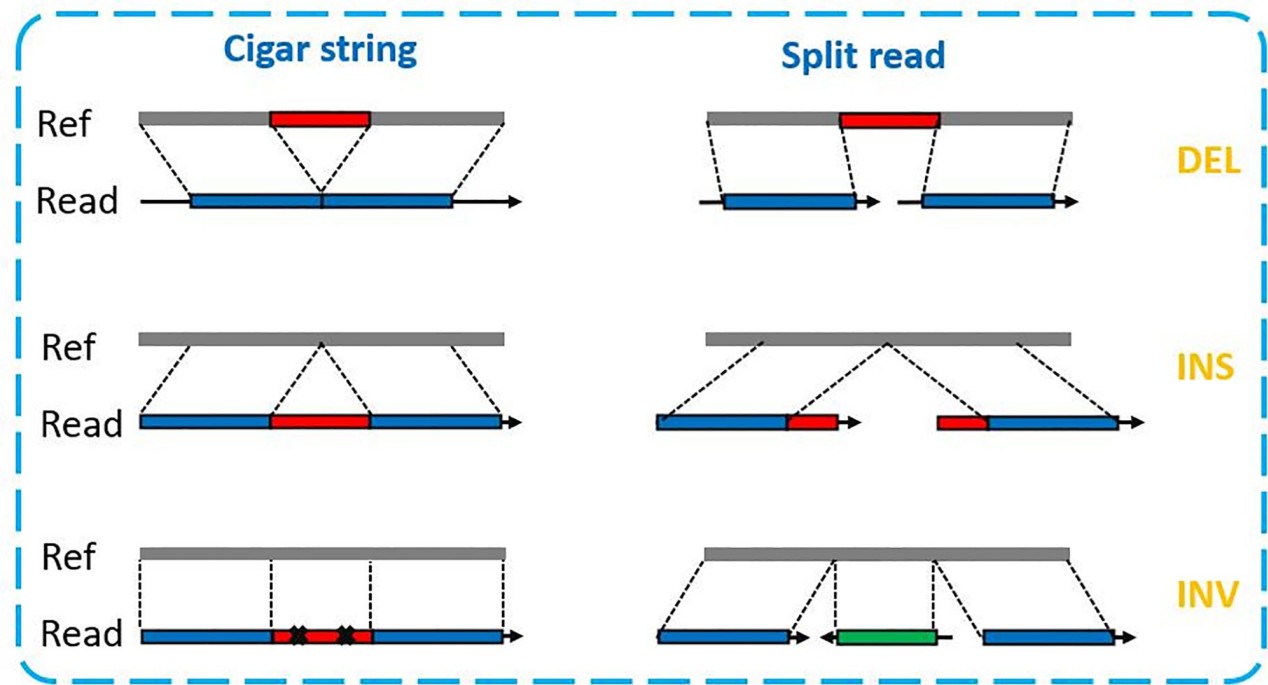

**Fig 3. On the left side, support reads with only one alignment for different types of SVs are shown.** We examine the cigar string for these reads and determine whether they are support reads. On the right side, support reads with two or three alignments are shown for different types of SVs. In such cases, we examine the split read and determine whether the read is a support read. The left-side type of support reads is more common for short SVs, while the right-side type is more common for long SVs.

alignment split from the BAM file can be collected. SVvalidation utilizes various methods to determine if a read supports the SV in question.

For the DEL or INS from the bed file (labeled as $indel_i$), we check whether the read has two alignments and whether the two alignments are on the same chromosome and on the same strand. Next, we calculate the distance of two alignments on reference and read and get two distances (ref_distance and read_distance). Then, we calculate the $difference_distances =$ read_distance-ref_distance. If difference_distances is greater than 200, the read supports an INS. Conversely, if difference_distances is less than -200, this read supports a DEL. If abs(abs (difference_distances)-$indel_i$_length) $\leq$ distance_support, we consider the read supports $indel_i$. The distance_support formula is the same as in the last section.

For the INV from the bed file (labeled as $INV_i$), we check whether the read has three alignments and whether these alignments are on the same chromosome and have different strands. If the read has three alignments (just like the split read of INV in Fig 3), we record the distance between the first alignment and three alignments (blue part in figure) and compare the distance with $INV_i$_length. If abs(distance-$INV_i$_length) $\leq$ distance_support, we consider the read supports $INV_i$.

If the INV is too large, the above way is not performing well. Because there is no read that can go through the entire INV region. For these reads, they only have two different strand alignments. If the read goes through the INV left breakpoint, we will record the left blue part end position and middle green part end position (split read of INV in Fig 3) and calculate the distance between the two positions. If the read goes through the INV right breakpoint, we will record the middle green part start position and right middle part start position (split

read of INV in Fig 3) and calculate the distance between the two positions. If abs(distance-$INV_i$_length) $\leq$ distance_support, we consider the read supports $INV_i$.

**Validating and outputting correct SVs.** We obtain the support read number and total read number (represented by the blue reads in step 2 of Fig 1) for each SV from the aforementioned steps. By applying the formula: support_rate = (support read number) / (total read number), we determine the type of the SV and report it as output.

$$support\_rate = \begin{cases} 0.8 - 1.0 & Homozygous, \\ 0.1 - 0.8 & Heterozygote, \\ 0 - 0.1 & false \end{cases}$$

If we judge that an SV from the bed file is a false SV and more than 10% of the reads in this region support another SV. This means that the previous SV is incorrect. We will calculate the average length from the reads support SV (blue reads in step 3 of Fig 1) and output the average length as the correct SV.

## Results

We evaluated the performance of SVvalidation and vapor by using the long-read datasets of HG002 and CHM13. The sequencing depths and links to download these datasets are provided in Table S1 of the S1 File. To assess the performance of different methods, we used recall, precision, and F1-score measurements. Recall was determined by calculating how many true SVs were correctly identified as such, while precision was calculated by determining how many SVs identified as true were indeed true SVs. Detailed results for the two methods are included in Tables S2–S4 of the S1 File. In addition, we compared the performance of SVvalidation with that of three other SV callers (cuteSV, Sniffles2, and DeBreak) and found that SVvalidation had superior performance.

### False negatives about vapor

This section aims to investigate why some true SVs cannot be validated by existing methods. When benchmarking HG002, vapor was only able to validate approximately 75% of true DEL and 76% of true INS in ONT long-read data. We examined the true SVs that Vapor failed to validate and discovered that over 90% of them were in repeat regions (instructions for downloading these regions can be found in the S1 File section 3). In contrast, only 30% of SVs successfully validated were in repeat regions. Therefore, we divided the benchmark into two categories: SVs in repeat regions and those in normal regions. The validated ratios are presented in Fig 4(a).

From the figure, it can be observed that vapor performs well in normal regions but only validates 49.5% of DELs and 52.4% of INSs in repeat regions. The reason behind this is that vapor generates a recurrence matrix by sliding a fixed-size window (k-mer) with a 1bp step to identify positions where the read sequence and reference match. However, it becomes challenging to pinpoint the accurate positions of sequences in the window in repeat regions. Further explanation for this is illustrated in Fig 5, where a red region represents a DEL in the repeat region. Taking a small sequence from the sample sequence (resembling the window portion in the figure), it can be noted that this small sequence has at least four positions to align. Therefore, vapor has relatively poor performance in repeat regions.

To address this issue, we propose a simple method. For the short sequences (just like the sequence in the window in Fig 5) in repeat regions, it is hard to get the correct alignment

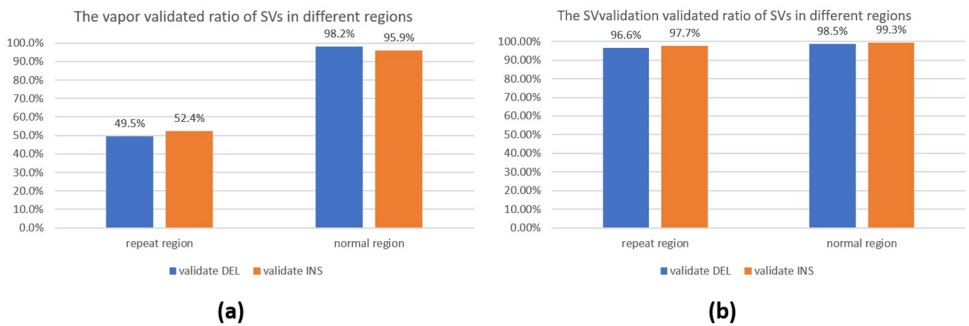

**Fig 4.** The bar chart (a) illustrates the ratio of SVs validated by vapor in repeat regions versus normal regions. Our conclusion is that vapor performs well in validating SVs in normal regions, but poorly in repeat regions. On the other hand, bar chart (b) depicts the SVvalidation-validated ratio of SVs in repeat and normal regions, indicating that SVvalidation has a better performance in validating SVs in repeat regions than vapor does.

location. However, if we use a longer read (the read can go through the whole repeat region) to validate, this problem will be solved. Consequently, we filter out reads that do not cover the repeat region and utilize the remaining reads to validate structural variations (SVs). The corresponding results are illustrated in Fig 4(b). It is notable that our proposed method—SVvalidation exhibits impressive performance with a validation ratio of over 96% for both normal and repeat regions, which demonstrates its superior effectiveness compared with the result presented in Fig 4(a).

## False positives of vapor

The previous section explained why vapor cannot validate some true SVs. In this section, we will demonstrate why existing methods produce false SVs. To test the ability of existing methods to filter out false SVs, we created a list of false SVs. Existing methods performed well with false SVs in the normal region (regions without any SVs). Thus, we generated a list of false SVs that are difficult to filter. We selected a subset of SVs from the benchmark and randomly assigned a number from [0.1, 0.2, 0.3, 0.4, 2, 3, 4, 5] to each SV. Then, we adjusted the length of each SV according to the assigned number. For example, if the original length of an SV was 500bp and the assigned number was 2, we created a false SV with a length of 1000bp (500*2). Subsequently, we obtained a list of SVs with problematic lengths with errors greater than 50%. Refer to the S1 File section 5 for detailed information. We applied vapor to validate the list of

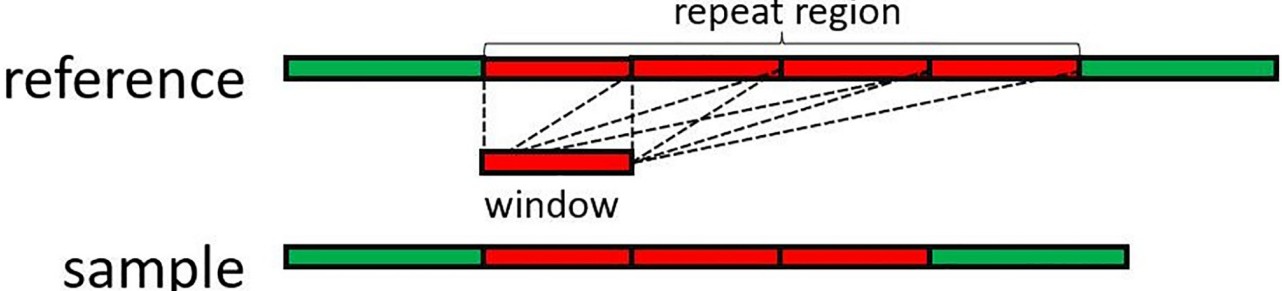

**Fig 5. In this figure, the red part is a repeat region and the sample has a DEL in this repeat region compared with the reference.** If we take out a short sequence by window, the short sequence has at least four positions to align.

false SVs. We discovered that vapor produced over 30% false DELs and 5% INSs. Based on these observations, we conclude that vapor is unsuitable for validating SVs with erroneous lengths.

## Performance in real data

Firstly, we obtained benchmarks for HG002 and CHM13. The HG002 benchmark includes 5463 deletions (DELs) and 7279 insertions (INSs), while the CHM13 benchmark includes 3647 DELs and 6457 INSs. Although these benchmarks are not comprehensive—studies have found that, on average, each human individual has around 20,000 structural variants—they boast incredibly high accuracy and thus can be considered as representing 100% true SVs. Next, we evaluated the validation performance of SVvalidation and Vapor using ONT long-read data. S1 File Table S1 provides a summary of sequencing depths and links to download the datasets. In addition to the benchmarks, we created two lists of false SVs for HG002 and CHM13, with the same number of entries as those in the corresponding benchmarks. We used these false SVs lists to assess the methods' abilities to filter out incorrect SV calls. More detailed results may be found in the S1 File Section 2. We did not present results for inversions as there is currently no high-confidence INV benchmark available and no perfect INV caller method exists. As such, only the results for DEL and INS were shown.

As shown in Fig 6, SVvalidation has the highest recall, precision, and F1-score among all the datasets. The recall value for SVvalidation is above 94% in all datasets, and the precision values in all datasets are above 87%. In comparison to vapor, our validation methodology results in an improvement of 7–16% for the F1-score.

## Performance in different coverage

To investigate the effect of sequencing depth, we randomly selected 30x, 20x, 10x, and 5x long reads from the ONT HG002 dataset and aligned them to GRCh37. This resulted in several new bam files, in which we applied the two methods and evaluated their performance at different coverage levels. Fig 7 portrays the F1-scores of each method at varying sequencing depths. Our results indicate that SVvalidation consistently outperforms other methods across all depths. Notably, even when the sequencing depth was only 5x, SVvalidation's F1-score exceeded 0.9.

Based on our research, we have found that the F1-score of SVvalidation is highest when coverage is at 30x. However, interestingly we observed that when coverage is at 5x, the F1-score for INS validation is actually higher than when it is at 10x. This can be attributed to the fact that low-coverage data results in a reduction of both true and false SVs that can be validated. When there is a decrease in coverage, the recall value tends to decrease while the precision increases. Nevertheless, SVvalidation generally performs better on high-coverage data.

## Performance of different callers

Compared with a few validation methods, researchers have developed many SV caller methods. Some SV callers include validation steps, though it is difficult to extract these steps individually in order to validate specific SVs listed. To address this challenge, we employed the following method to compare SVvalidation with other SV callers based on their validation performance: First, we ran each SV caller using long-read data from HG002 and CHM13, then compared the output against the benchmark for those datasets. If an SV caller includes SVs that appear in the benchmark sets, it can be considered successfully validated (For more information about how to check whether an SV caller outputs corresponding SVs identified in the benchmark, please refer to the S1 File Section 4).

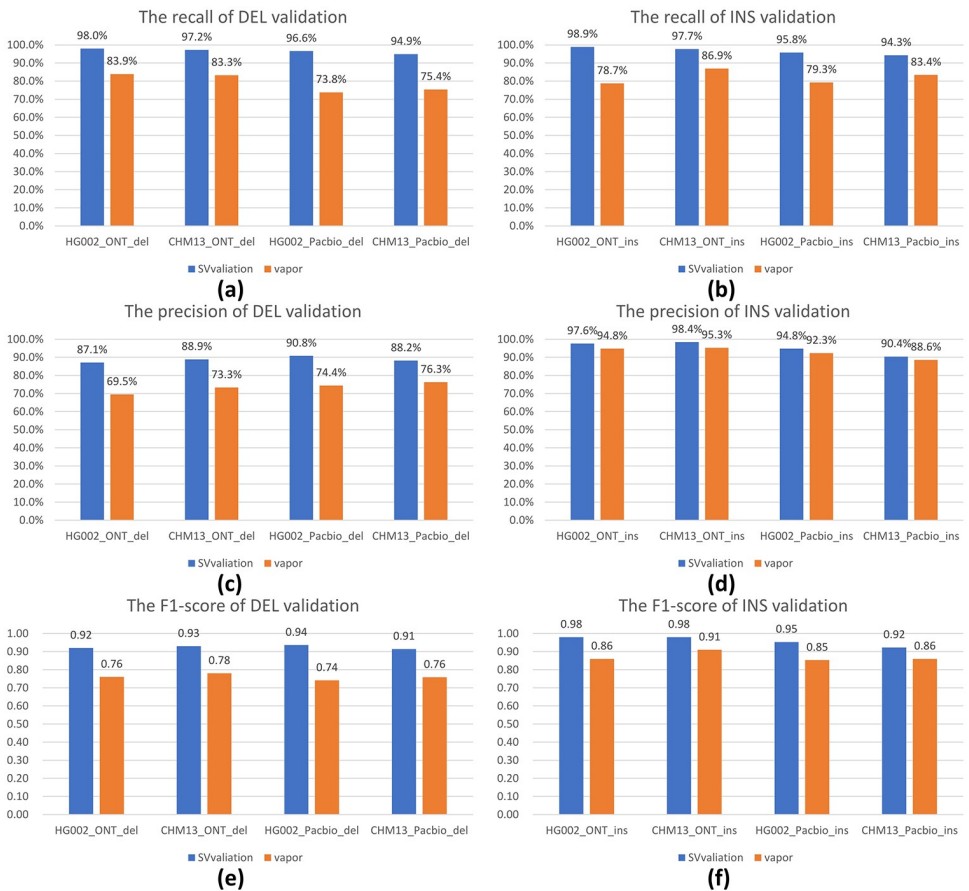

**Fig 6. These histograms show the recall, precision, and F1-score of different methods validation on HG002 and CHM13 in different long-read data. (a)** The recall of DELs validation for HG002 and CHM13 in ONT and pacbio data. **(b)** The recall of INSs validation for HG002 and CHM13 in ONT and pacbio data. **(c)** The precision of DELs validation for HG002 and CHM13 in ONT and pacbio data. **(d)** The precision of INSs validation for HG002 and CHM13 in ONT and pacbio data. **(e)** The F1-score of DELs validation for HG002 and CHM13 in ONT and pacbio data. **(f)** The F1-score of INSs validation for HG002 and CHM13 in ONT and pacbio data. As shown in the figure, we can see that SVvalidation can achieve the best results in the recall, precision, and F1-score.

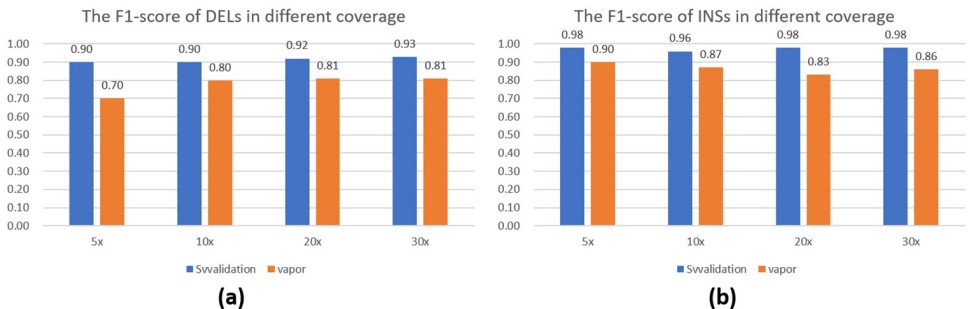

**Fig 7.** (a) The F1-score of SVvalidation and vapor DEL validation in HG002 under different sequencing depths. (b) The F1-score of SVvalidation and vapor INS validation in HG002 under different sequencing depths. In the figure, SVvalidation has the best performance in all coverage datasets.

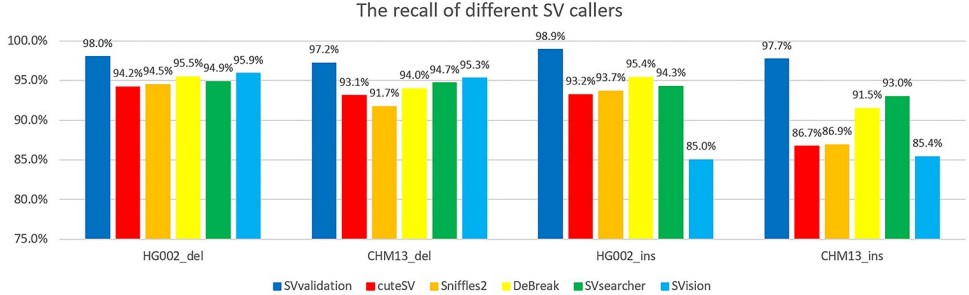

**Fig 8. These histograms display the recall values of different SV callers on HG002 and CHM13 samples.** As illustrated in the figure, it is evident that the recall value of SVvalidation is greater than current SV callers.

Due to the impracticality of testing all long-read SV callers, we selected five methods—cuteSV (version 2.0.2), sniffles2 (version 2.0), DeBreak (latest version), SVsearcher (latest version) and SVision (latest version)—based on their popularity and high accuracy. We ran five methods on ONT long-read datasets of HG002 and CHM13 (download links provided in the S1 File Table S1). We then compared their results to benchmarks and calculated the number of true SVs found by each method (recorded as find_trueSV_num). Lastly, we determined the recall value using the formula: recall = find_trueSV_num/benchmark_SV_num.

Fig 8 displays the recall values for cuteSV, Sniffles2, and DeBreak in the HG002 and CHM13 datasets. As shown in the figure, SVvalidation still has the highest recall value since SV callers first need to detect SVs and then validate them. Nonetheless, cuteSV, Sniffles2, and DeBreak exhibit good performance in identifying SVs as they can produce more than 90% of SVs in the benchmark. We did not include precision in our benchmark as it only includes high confidence but incomplete results. Studies suggest that each individual has an average of 20,000 structural variants [35]. Our benchmark, however, comprises only 12742 SVs in the HG002 dataset, and we cannot classify the SVs that are not in the benchmark as false SVs. Nevertheless, cuteSV and Sniffles2 both output more than 29,000 SVs, whereas DeBreak outputs more than 23,000 SVs. Therefore, assuming an individual contains about 20,000 SVs, cuteSV and Sniffles2 yield at least 9,000 false SVs, while DeBreak produces at least 3,000 false SVs.

## The running time

We used the HG002 ONT dataset with a sequencing depth 50x to run SVvalidation, vapor, cuteSV, Sniffles2, and DeBreak, and recorded the running time for each tool. The details are presented in Table 1.

## Conclusion

In previous studies, we have discovered that while there are numerous methods for detecting SVs, very few of these methods have been developed for validating SVs. Upon reviewing current validation methods, we found that their performance was subpar. To tackle this issue, we propose a new method called SVvalidation. Given the higher reliability of long read alignments

**Table 1. The run time of different methods.**

| Method | SVvalidation | vapor | cuteSV | Sniffles2 | DeBreak |
|---|---|---|---|---|---|
| time | 37m | 6h27m | 16m | 20m | 48m |

compared to short read, we will leverage long read data for SV validation. SVvalidation has several advantages over existing methods, which include:

1. SVvalidation can validate almost true SVs (including those in repeat regions).

2. SVvalidation can filtrate the false SVs (include the SVs with error length) and output correct SVs.

3. SVvalidation has the best performance than other methods in different coverage.

Despite its efficacy, SVvalidation still has limitations when dealing with complex SVs (two different types of SVs that are adjacent). Also, certain regions in the genome have high similarities, resulting in most long-read alignments having a MAPQ of 0, making it challenging to identify high-quality alignments for validating SVs. Consequently, our method may not accurately validate SVs in these regions. Our future research will focus on exploring additional strategies to validate complex SVs.

## Supporting information

**S1 File. This file contains the data sources and program commands and detailed step of some methods.**
(PDF)

## Acknowledgments

We would like to thank the helpful discussion with Professor Wing-Kin Sung in the computing school of NUS.

## Author Contributions

**Data curation:** Yan Zheng.

**Funding acquisition:** Xuequn Shang.

**Software:** Yan Zheng.

**Validation:** Yan Zheng.

**Writing – original draft:** Yan Zheng.

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
