## [Decision Letter · Decision Letter 0]

20 Aug 2023

PONE-D-23-22329SVvalidation: a long-read-based validation method for genomic structural variationPLOS ONE

Dear Dr. Zheng,

Thank you for submitting your manuscript to PLOS ONE. After careful consideration, we feel that it has merit but does not fully meet PLOS ONE’s publication criteria as it currently stands. Therefore, we invite you to submit a revised version of the manuscript that addresses the points raised during the review process.

We look forward to receiving your revised manuscript.

Kind regards,

Zechen Chong

Academic Editor

PLOS ONE

Journal Requirements:

a) The name of the colleague or the details of the professional service that edited your manuscript.

b) A copy of your manuscript showing your changes by either highlighting them or using track changes (uploaded as a *supporting information* file).

c) A clean copy of the edited manuscript (uploaded as the new *manuscript* file).

3. Thank you for stating the following financial disclosure: "The author(s) received no specific funding for this work."

5. Please update your submission to use the PLOS LaTeX template. The template and more information on our requirements for LaTeX submissions can be found at http://journals.plos.org/plosone/s/latex.

**Additional Editor Comments:**

The authors should address Reviewer 1 and Reviewer 2's major concerns. Especially, the authors should demonstrate SVvalidation's performance on other types of SVs on a diverse datasets, a detailed method should be provided, and the source code should be made accessible.

Reviewers' comments:

Reviewer's Responses to Questions

**Comments to the Author**

1. Is the manuscript technically sound, and do the data support the conclusions?

Reviewer #1: Yes

Reviewer #2: Yes

2. Has the statistical analysis been performed appropriately and rigorously? 

Reviewer #1: Yes

Reviewer #2: N/A

3. Have the authors made all data underlying the findings in their manuscript fully available?

Reviewer #1: Yes

Reviewer #2: No

4. Is the manuscript presented in an intelligible fashion and written in standard English?

Reviewer #1: No

Reviewer #2: Yes

5. Review Comments to the Author

Reviewer #1: The authors proposed a long read-based structural variation validation method called SVvalidation, which takes BED and BAM files as inputs, and filters structural variants within BED by estimating parameters, identifying supported reads, and validating structural variations, in order to output higher-quality detection results. The experimental results show that SVvalidation has higher F1 scores for INSs and DELs compared to existing methods vapor, cuteSV, Sniffles and DeBreak.

To further improve the quality of the manuscript, I have some concerns as follows:

1."Structural Variations (SVs) [1] refer to large-scale mutations in a genome, including deletions, duplications, inversions, and translocations, with a length of at least 50 base pairs."

As far as I know, and as described in the reference [1], structural variation should also include insertion variation (INS).

2.In recent years, some new long read-based structural variant detection algorithms have been published:

(1) SVision: a deep learning approach to resolve complex structural variants. Nature Methods, 2022, 19(10): 1230-1233.

(2) MAMnet: detecting and genotyping deletions and insertions based on long reads and a deep learning approach. Briefings in Bioinformatics, 2022, 23(5): bbac195.

(3) cnnLSV: detecting structural variants by encoding long-read alignment information and convolutional neural network. BMC Bioinformatics, 2023, 24(1): 1-19.

(4) SVsearcher: A more accurate structural variation detection method in long read data. Computers in Biology and Medicine, 2023, 158: 106843.

3."Additionally, it is noteworthy that SVvalidation is capable of accurately validating all types of SVs. "

However, the experimental results demonstrated the outstanding performance of SVvalidation on only INSs and DELs, but not including other types of SVs, for example INVs, DUPs, or TRAs.

If your dataset used was limited in the experiment, simulated dataset or other real dataset like NA19240 can be used.

4.It is recommended that the authors give a formal description and analysis of the algorithm SVvalidation.

5.The source code address provided in the manuscript is not accessible. The actual accessible address on github is https://github.com/nwpuzhengyan/SVvaliation. Please check and fix it.

6.Please check and correct grammatical errors in the manuscript and improve the presentation (e.g., set parameters in part “Identifying support reads” should be written in italics).

Reviewer #2: Comments to "SVvalidation: a long-read-based validation method for genomic structural variation"

The authors proposed a novel method called SVvalidation which validates SVs with higher accuracy using long reads sample on Genome in a Bottle (HG002 and CHM13), determines the correct length of SVs, and outputs the precise SVs compared to existing methods. The rationale states that SV callers produce many false positive SVs, and existing validation methods are not accurate enough, especially in repetitive regions. The authors construct three steps to validate SVs: (1) estimating parameters, (2) identifying supported reads, and (3) validating and producing accurate SVs output. This allows them to validate almost true SVs (include the SVs in repeat regions), filtrate the false SVs (include the SVs with error length) and output correct SVs, achieve higher accuracy, and better performance than other tools in different coverage. The topic is important and demanding, especially in repetitive and complex regions. The solution seems working, but the evidence is not efficient. I have the following concerns:

1. This paper employs a filter to identify repetitive regions by discarding SVs occurring in regions with coverage exceeding five times the average. Is there substantial evidence supporting the selection of this specific five-fold cutoff for repetitive regions?

2. Have the authors validated the performance of SVvalidation on a diverse set of samples to ensure consistent sensitivity and precision without overfitting?

3. Considering that benchmark high confidence regions typically not include repetitive regions, is there a well-defined gold standard for SVs in repetitive regions within the genome? If not, how did the authors establish a standard for SV comparison when using SVvalidation?

4. The authors claim that SVvalidation is applicable to all datasets. However, is its accuracy also high when applied to short reads, or is it particularly and primarily optimized for long-read data?

5. I notice the program is not available on GitHub. It would be beneficial for the availability of the program or sharing it with the journal for review purposes, allowing reviewers to evaluate the effectiveness of this method.

6. PLOS authors have the option to publish the peer review history of their article (what does this mean?). If published, this will include your full peer review and any attached files.

Reviewer #1: No

Reviewer #2: **Yes: **Kaili Hu

---

## [Author Response · Author response to Decision Letter 0]

1 Sep 2023

Dear Editor and Reviewers,

Thank you for your decision and constructive comments on my manuscript. We have carefully considered the suggestion of Reviewer and make some changes. We have tried our best to improve and made some changes in the manuscript. The red parts in manuscript have been revised according to your comments. Replies to reviewers are given as follows:

Reviewer #1: The authors proposed a long read-based structural variation validation method called SVvalidation, which takes BED and BAM files as inputs, and filters structural variants within BED by estimating parameters, identifying supported reads, and validating structural variations, in order to output higher-quality detection results. The experimental results show that SVvalidation has higher F1 scores for INSs and DELs compared to existing methods vapor, cuteSV, Sniffles and DeBreak. 

To further improve the quality of the manuscript, I have some concerns as follows: 

1."Structural Variations (SVs) [1] refer to large-scale mutations in a genome, including deletions, duplications, inversions, and translocations, with a length of at least 50 base pairs." As far as I know, and as described in the reference [1], structural variation should also include insertion variation (INS).

Reply to reviewer: Thank you for your suggestion. We have added insertions into this sentence.

2.In recent years, some new long read-based structural variant detection algorithms have been published: (1) SVision: a deep learning approach to resolve complex structural variants. Nature Methods, 2022, 19(10): 1230-1233. (2) MAMnet: detecting and genotyping deletions and insertions based on long reads and a deep learning approach. Briefings in Bioinformatics, 2022, 23(5): bbac195. (3) cnnLSV: detecting structural variants by encoding long-read alignment information and convolutional neural network. BMC Bioinformatics, 2023, 24(1): 1-19. (4) SVsearcher: A more accurate structural variation detection method in long read data. Computers in Biology and Medicine, 2023, 158: 106843. 

Reply to reviewer: Thank you for your suggestion. We have added SVision and SVsearcher result into the paper.

3."Additionally, it is noteworthy that SVvalidation is capable of accurately validating all types of SVs. " However, the experimental results demonstrated the outstanding performance of SVvalidation on only INSs and DELs, but not including other types of SVs, for example INVs, DUPs, or TRAs. If your dataset used was limited in the experiment, simulated dataset or other real dataset like NA19240 can be used. 

Reply to reviewer: Thank you for your suggestion. Because we consider DUP as a special type of INS, the validation of DUP is similar to that of INS, and we did not show it separately. For TRAs, our method has not implemented the function of validating TRAs. For INVs, we have created simulated INVs by reversing the sequence of reference. However, the validation methods all performed very well on simulated INVs. Therefore, we don't think this result is meaningful, so we don’t show it.

4.It is recommended that the authors give a formal description and analysis of the algorithm SVvalidation. 

Reply to reviewer: Thank you for your suggestion. We have revised our expression to make the paper easier to understand.

5.The source code address provided in the manuscript is not accessible. The actual accessible address on github is https://github.com/nwpuzhengyan/SVvaliation. Please check and fix it. 

Reply to reviewer: Thank you for your suggestion. We have fixed my source code address.

6.Please check and correct grammatical errors in the manuscript and improve the presentation (e.g., set parameters in part “Identifying support reads” should be written in italics).

Reply to reviewer: Thank you for your suggestion. We have corrected our grammar and improved the presentation.

Reviewer #2: Comments to "SVvalidation: a long-read-based validation method for genomic structural variation". The authors proposed a novel method called SVvalidation which validates SVs with higher accuracy using long reads sample on Genome in a Bottle (HG002 and CHM13), determines the correct length of SVs, and outputs the precise SVs compared to existing methods. The rationale states that SV callers produce many false positive SVs, and existing validation methods are not accurate enough, especially in repetitive regions. The authors construct three steps to validate SVs: (1) estimating parameters, (2) identifying supported reads, and (3) validating and producing accurate SVs output. This allows them to validate almost true SVs (include the SVs in repeat regions), filtrate the false SVs (include the SVs with error length) and output correct SVs, achieve higher accuracy, and better performance than other tools in different coverage. The topic is important and demanding, especially in repetitive and complex regions. The solution seems working, but the evidence is not efficient. I have the following concerns: 

1. This paper employs a filter to identify repetitive regions by discarding SVs occurring in regions with coverage exceeding five times the average. Is there substantial evidence supporting the selection of this specific five-fold cutoff for repetitive regions? 

Reply to reviewer: Thank you for your comment. The SVs in these abnormally high coverage regions are only a small fraction of the actual number. In the CHM13 benchmark, only 0.5% of DEL and 1% of INS are in the regions where the coverage exceeds five times the average coverage. This has little effect on the results of the validation.

2. Have the authors validated the performance of SVvalidation on a diverse set of samples to ensure consistent sensitivity and precision without overfitting? 

Reply to reviewer: We have run our program on two datasets, CHM13 and HG002, and each dataset has been run on both ONT and PACBIO long reads. The results show that our method indeed performs well. So we believe that our program does not have overfitting.

3. Considering that benchmark high confidence regions typically not include repetitive regions, is there a well-defined gold standard for SVs in repetitive regions within the genome? If not, how did the authors establish a standard for SV comparison when using SVvalidation? 

Reply to reviewer: Thank you for your comment. We downloaded the repeat regions of the reference from Table Browser. If 80% of an SV is in the repeat region, we consider it to be an SV in the repeat region. The details are shown in the supplementary file section 3.

4. The authors claim that SVvalidation is applicable to all datasets. However, is its accuracy also high when applied to short reads, or is it particularly and primarily optimized for long-read data? 

Reply to reviewer: My method only applies to long reads, not short reads.

5. I notice the program is not available on GitHub. It would be beneficial for the availability of the program or sharing it with the journal for review purposes, allowing reviewers to evaluate the effectiveness of this method.

Reply to reviewer: Thank you for your suggestion. We have fixed my source code address. The address is now available.

We are extremely grateful to Reviewers for pointing out these problems. We have revised these problems one by one.

Thank you and all the reviewers for the kind advice again. If you have any questions, please contact us without hesitate.

Yours sincerely,

Yan Zheng and Xuequn Shang

---

## [Editor Report · Decision Letter 1]

6 Sep 2023

SVvalidation: a long-read-based validation method for genomic structural variation

PONE-D-23-22329R1

Dear Dr. Zheng,

We’re pleased to inform you that your manuscript has been judged scientifically suitable for publication and will be formally accepted for publication once it meets all outstanding technical requirements.

Kind regards,

Zechen Chong

Academic Editor

PLOS ONE
---

## [Editor Report · Acceptance letter]

12 Sep 2023

PONE-D-23-22329R1 

SVvalidation: a long-read-based validation method for genomic structural variation 

Dear Dr. Zheng:

I'm pleased to inform you that your manuscript has been deemed suitable for publication in PLOS ONE. Congratulations! Your manuscript is now with our production department. 

Kind regards, 

on behalf of

Dr. Zechen Chong 

Academic Editor

PLOS ONE